# The Systemic Immune Response in COVID-19 Is Associated with a Shift to Formyl-Peptide Unresponsive Eosinophils

**DOI:** 10.3390/cells10051109

**Published:** 2021-05-05

**Authors:** Leo Koenderman, Maarten J. Siemers, Corneli van Aalst, Suzanne H. Bongers, Roy Spijkerman, Bas J. J. Bindels, Giulio Giustarini, Harriët M. R. van Goor, Karin A. H. Kaasjager, Nienke Vrisekoop

**Affiliations:** 1Department of Respiratory Medicine, University Medical Center Utrecht, 3584CX Utrecht, The Netherlands; m.j.siemers@students.uu.nl (M.J.S.); c.vanaalst@umcutrecht.nl (C.v.A.); r.spijkerman@umcutrecht.nl (R.S.); b.j.j.bindels@umcutrecht.nl (B.J.J.B.); G.Giustarini@umcutrecht.nl (G.G.); n.vrisekoop@umcutrecht.nl (N.V.); 2Center for Translational Immunology, University Medical Center Utrecht, 3584CX Utrecht, The Netherlands; s.h.bongers@umcutrecht.nl; 3Department of Trauma Surgery, University Medical Center Utrecht, 3584CX Utrecht, The Netherlands; 4Department of Internal Medicine, University Medical Center Utrecht, 3584CX Utrecht, The Netherlands; H.M.R.vanGoor-3@umcutrecht.nl (H.M.R.v.G.); h.a.h.kaasjager@umcutrecht.nl (K.A.H.K.)

**Keywords:** SARS-CoV-2, COVID-19, point-of-care flow cytometry, eosinophil, formyl peptide, responsiveness, blood

## Abstract

A malfunction of the innate immune response in COVID-19 is associated with eosinopenia, particularly in more severe cases. This study tested the hypothesis that this eosinopenia is COVID-19 specific and is associated with systemic activation of eosinophils. Blood of 15 healthy controls and 75 adult patients with suspected COVID-19 at the ER were included before PCR testing and analyzed by point-of-care automated flow cytometry (CD10, CD11b, CD16, and CD62L) in the absence or presence of a formyl peptide (fNLF). Forty-five SARS-CoV-2 PCR positive patients were grouped based on disease severity. PCR negative patients with proven bacterial (*n* = 20) or other viral (*n* = 10) infections were used as disease controls. Eosinophils were identified with the use of the FlowSOM algorithm. Low blood eosinophil numbers (<100 cells/μL; *p* < 0.005) were found both in patients with COVID-19 and with other infectious diseases, albeit less pronounced. Two discrete eosinophil populations were identified in healthy controls both before and after activation with fNLF based on the expression of CD11b. Before activation, the CD11b^bright^ population consisted of 5.4% (CI_95%_ = 3.8, 13.4) of total eosinophils. After activation, this population of CD11b^bright^ cells comprised nearly half the population (42.21%, CI_95%_ = 35.9, 54.1). Eosinophils in COVID-19 had a similar percentage of CD11b^bright^ cells before activation (7.6%, CI_95%_ = 4.5, 13.6), but were clearly refractory to activation with fNLF as a much lower percentage of cells end up in the CD11b^bright^ fraction after activation (23.7%, CI_95%_ = 18.5, 27.6; *p* < 0.001). Low eosinophil numbers in COVID-19 are associated with refractoriness in responsiveness to fNLF. This might be caused by migration of fully functional cells to the tissue.

## 1. Introduction

Coronavirus disease 2019 (COVID-19), a rapidly emerging pandemic disease caused by the Severe Acute Respiratory Syndrome Coronavirus 2 (SARS-CoV-2), causes an extremely variable disease ranging from asymptomatic to viral pneumonia, respiratory distress, ICU admission, and/or death [1,2]. There are still many questions regarding the dominant pathogenetic mechanisms underlying the cause and course of disease [2]. Several hypotheses have been put forward and two of these that are not mutually exclusive have received a lot of attention in the recent literature: coagulopathy associated with tissue edema and (micro)thrombosis [3] and hyperactivation of the innate immune system [4] associated with a cytokine storm [5].

Clinically severe to critical COVID-19 in the ICU has many characteristics of coagulopathy and tissue edema, particularly in the lung [6]. This supports the concept that ARDS in COVID-19 is mainly caused by tissue edema that is mediated by a deregulated kininogen–bradykinin pathway [3]. Supporting evidence comes from the fact that the SARS-CoV2 virus uses angiotensin-converting enzyme-2 (ACE-2) as the front door in epithelial cells, which is associated with the downregulation of ACE-2 from the cell surface [2]. A decrease in ACE-2 is thought to be involved in the attenuated breakdown of bradykinin [3], leading to tissue edema in the lung and a clinical picture reminiscent of ARDS [6]. Despite its significance, coagulopathy is beyond the scope of this article, which is focused on a putative dysfunction of the innate immune response [4].

A multitude of articles describe the importance of hyperinflammation in the pathogenesis of COVID-19 (see for recent reviews [2,5,7]). Most of these studies had a case control design using healthy individuals as controls. These studies generally show more systemic inflammation with increasing disease severity. This is associated with a cytokine storm and clear differences in the phenotype and/or activation status of the different innate immune cells [8]. Deep phenotyping has identified several neutrophil phenotypes leading to the conclusion that hyperinflammation is a specific and integral part of the pathogenesis of COVID-19. However, recent studies refute the importance of a cytokine storm and neutrophil activation in the pathogenesis of COVID-19 at least in the population of patients outside the intensive care unit. In fact, these studies imply that these mechanisms are commonly initiated in any severe/critical inflammatory disease and are not COVID-19 specific [1,8,9].

The importance of the eosinophilic compartment in the pathogenesis of COVID-19 is less clear. Early studies, particularly those coming from Wuhan, have indicated that more severe COVID-19 is associated with eosinopenia [10,11]. Unfortunately, no underlying mechanism was identified. Furthermore, the activation status of the remaining eosinophils in the peripheral blood was not studied.

Here, we set out experiments to study the eosinophil compartment in COVID-19 patients both qualitatively and quantitatively. We will show also that our COVID-19 cohort was associated with a relative eosinopenia, but eosinopenia was also seen in some patients with other viral and bacterial infections, albeit to a lower extent. In addition, the eosinophil compartment exhibited clear signs of deregulated activation. Whether this activation is pathological or a normal response to virus infections will be discussed.

## 2. Materials and Methods

### 2.1. Study Design

Seventy-five suspected COVID-19 patients older than 18 years of age were included who presented at the University Medical Center Utrecht (UMCU) during the Dutch epidemic between March 19th 2020 and May 17th 2020. SARS-CoV2 PCR positive patients (*n* = 45) were compared to healthy controls (*n* = 15) and SARS-CoV2 PCR negative patients with an alternative diagnosis (viral (*n* = 10) or bacterial infection (*n* = 20)). The initial blood samples were drawn at the emergency room (ER) or within two days on the COVID-19 ward. Thereafter, samples were drawn at regular intervals during the stay at the COVID-19 ward. No blood samples were drawn at the intensive care unit. Clinical outcomes and baseline characteristics of the COVID-19 patients have been published recently [8]. The patients were not immunocompromised nor used immunosuppressive medication according to the International Classification of Disease, 9th revision [12].

### 2.2. Study Procedure

Blood was drawn in one extra 4 or 9 mL VACUETTE^®^ sodium heparin blood tube (Greiner Bio-One, Kremsmünster, Austria) from each patient during standard-of-care diagnostic workup at the Emergency Department (the formation of a COVID-19-biobank (#20-175) with a reviewed issuance protocol was approved on the 21 of April 2020) or during standard blood drawing moments at the COVID-19 ward. Healthy control blood was obtained from the Mini Donor Service at the UMCU (the Mini Donor Service received positive approval (already in 2007) from the medical ethical committee of the UMCU under protocol number 07-125/C. This service provides normal donors (voluntary hospital personnel) to donate blood for any research). Healthy control subjects could not be matched for age nor gender because of the limitations for clinical experiments during the first wave of the pandemic and had to be chosen based on donor availability. The blood from healthy controls was drawn at the out-patient clinic, which is next to the emergency department. Sample handling times analysis were identical to the patients included in this study. SARS-CoV-2 was detected with the Allplex 2019-nCoV assay (Seegene, Düsseldorf, Germany). Diagnostics of other viruses were performed with the ePlex RP panel (GenMarkDx, Carlsbad, CA, USA)

### 2.3. Flow Cytometry Analysis by the Automated AQUIOS CL^®^ “Load & Go” Flow Cytometer

Blood was analyzed in terms of total cell counts, scatter, and fluorescence by a completely automated point-of-care AQUIOS CL^®^ “Load & Go” flow cytometer (Beckman Coulter, Miami, FL, USA) exactly as described before [13]. Aspecific granulocyte activation in the blood tube ex vivo was minimized by applying a short (<60 min) time between blood drawing and analysis [13]. The analysis of the blood samples was exactly as described before [13]. In short, the whole blood samples were stained by a customized antibody mix (CD16-FITC (clone 3G8, Beckman Coulter, Miami, FL, USA), CD11b-PE (clone Bear1, Beckman Coulter), CD62L-ECD (clone DREG56, Beckman Coulter), CD10-PC5 (clone ALB1, Beckman Coulter), and CD64-PC7 (clone 22, Beckman Coulter)) in the absence (control) and presence (activation) of the bacterial/mitochondrial-derived stimulus N-formyl-norleucyl-leucyl-phenylalanine (10 μM fNLF) (BioCat GmbH, Heidelberg, Germany). Thereafter, the red blood cells were lysed, and the data were analyzed and exported as .lmd files for later in-depth analysis.

### 2.4. Analysis of Flow Cytometry Data

The .lmd data files obtained by AQUIOS CL^®^ were imported into the online tool Cytobank (www.cytobank.org, Santa Clara, CA, USA; accessed on 7 December 2020). The identification of eosinophils in the blood samples was conducted by gating first on granulocytes on the basis of scatter characteristics (see Figure 1). Thereafter, a second gate was set outside the majority of neutrophils (CD16^bright^) to enrich for eosinophils (CD16^negative^) and progenitors (CD16^dim^). Thereafter, this fraction was analyzed by the FlowSOM algorithm [14] (present in Cytobank), choosing 64 nodes and 6 metaclusters (Figure 1C). Next, the data were analyzed to establish that metacluster 3 contained the eosinophils (SSC^high^, CD16^negative^, and CD11b^moderate^) (Figure 1D,E). In this metacluster, the percentage and the fNLF responsiveness of the eosinophils present in the sample were determined. To this end, the eosinophil markers were analyzed in the absence (resting) and presence (activated) of fNLF (10 μM).

Two eosinophil phenotypes could be discriminated by either moderate or high expression of CD11b (see Figure 2). The marker CD64 was used to facilitate differentiation between viral and bacterial infections in COVID-19 negative patients, as has been demonstrated before [15]. CD64 was not applied for uni -and bivariate analysis of the eosinophil compartment.

### 2.5. Clinical Characteristics of Suspected COVID-19

Patients that tested PCR positive for SARS-CoV-2 at any point during admission were considered as having COVID-19. Patients that were PCR negative for COVID-19 were studied in detail for other diagnoses, explaining their clinical condition applying bacterial cultures, PCRs for other viruses, or other diagnostic tests. PCR negative patients with no other explaining diagnosis were excluded from the study.

Patients that were PCR positive for SARS-CoV-2 were grouped based on disease severity according WHO criteria published on 27th May 2020 (Clinical Management of COVID-19 [16]). Moderate disease was defined by the presence of mild to moderate pneumonia (including SpO2 ≥ 90% on room air). The definition of severe disease included severe pneumonia (including SpO2 < 90% on room air) and one of the following: respiratory rate > 30 breaths/min; severe respiratory distress, but without meeting the Berlin criteria [17] for ARDS at any time during admission. Critical disease was defined by the presence of ARDS SpO_2_/FiO_2_ ≤ 315 [16] at any point during admission.

### 2.6. Study Approval

For this study, the IRB approved the formation of a COVID-19-biobank (#20-175) with a reviewed issuance protocol. For this procedure, informed consent was waived. The Mini Donor Service received positive approval from the medical ethical committee of the UMCU under protocol number 07-125/C. All procedures performed in this study were in accordance with the 1964 Helsinki declaration and its later amendments.

### 2.7. Statistics

Kruskal–Wallis analysis was applied for the eosinophil data when multiple groups were compared. For comparison of CD11b^bright^ and CD11b^low^ cells, a Wilcoxon matched-pairs signed rank test was used. Healthy individuals and COVID-19 PCR negative patients with proven bacterial or other viral infection were used as separate control groups. Differences in receptor expression on eosinophils were tested with Kruskal–Wallis and post-hoc, and the Dunn’s test with Bonferroni correction was used for multiple comparison. GraphPad Prism version 7 (GraphPad software inc., San Diego, CA, USA) was used for analysis and data visualization. Statistical significance was defined as a *p*-value < 0.05.

## 3. Results

### 3.1. Responsiveness for fNLF Identifies Two Eosinophil Phenotypes in the Peripheral Blood of Healthy Control Individuals

Automated flowcytometry allows quick (25 min post venipuncture) determination of the phenotype of leukocytes with a minimal artificial activation due to ex vivo manipulation of the cells [13]. In fact, we have shown that expression of activation markers such as Mac-1/CD11b on eosinophils is quickly induced in the blood sample in the collection tube without any additional manipulation. This artificial activation is for a large part prevented by the fast analysis in the AQUIOS [13]. Applying this method, we identified two discrete populations of eosinophils in healthy control individuals: a dominant population that is characterized by a low expression of CD11b (CD11b^moderate^) and a much smaller population of CD11b^bright^ eosinophils (Figure 2).

Eosinophils have been described to be responsive to the formyl peptide formyl-methionyl-leucyl phenylalanine (fMLF [18]) and this response is sensitive for priming by inflammatory mediators [19]. Unfortunately, the methionyl moiety in fMLF is sensitive for oxidation that leads to the biological inactivity of the peptide. This precludes application in our automate flow analysis as the formyl peptide is present in the deep well plate as long as required for all the analyses. In marked contrast to fMLF, that loses its activity in days, fNLF is stable for weeks to months [8]. Activation of whole blood with fNLF (10 μM) leads to a marked increase in cells found in the CD11b^bright^ population (Figure 2). Interestingly, a sizeable number of eosinophils in the CD11b^moderate^ did not respond to fNLF (Figure 2). In line, CD62L was shed in the CD11b^bright^ but not in the CD11b^moderate^ population (Figure 2C). The small number of eosinophils with the CD11b^bright^ phenotype did not further upregulate CD11b in response to fNLF (Figure 2).

### 3.2. Eosinophil Numbers in the Peripheral Blood of COVID-19 Patients, Disease Controls, and Healthy Controls

Figure 3 shows the absolute eosinophil counts in the peripheral blood of the different categories of patients and controls at presentation in the hospital (panel A, see also [8]). Eosinophils were identified on the basis of metacluster 3, identified by the FlowSOM algorithm [14] (see Figure 1 and M&M section). In some experiments, this gate was validated by additional experiments using the eosinophil marker CCR3 (CD193) (results not shown). As shown before [8], COVID-19 patients were characterized by a relative eosinopenia but a relative eosinopenia was also found in the patients with other infections, albeit less pronounced (*p* = 0.02, Figure 3A). The eosinophil numbers normalized during improvement (see Figure 3B). At the end of the longitudinal individual patient trajectories (indicated with a patient specific color), the patients were discharged from the hospital.

### 3.3. Responsiveness of the Eosinophil Phenotypes for fNLF at Presentation in the Hospital: Comparison between COVID-19 Patients, Disease Controls, and Healthy Controls

The responsiveness for fNLF was determined for eosinophils from the blood of COVID-19 patients and the different control groups such as described above. In marked contrast to healthy controls, eosinophils in the blood of COVID-19 patients were refractory to activation with fNLF (see Figure 4). Interestingly, this refractoriness is not specific for COVID-19 as both bacterial and viral infections lead to a similar refractoriness of responsiveness to fNLF. Interestingly, the responsiveness of eosinophils for fNLF was irrespective of disease severity as the percentage of CD11b^bright^ cells in the absence or presence of the formyl peptide was similar in moderate, severe, and critical patients (see Figure 4C,D).

### 3.4. Normalization of Eosinophil Counts and Responsiveness to fNLF during Resolving Disease in the COVID-19 Ward

Next, we tested whether eosinophil counts and eosinophil responsiveness for fNLF would normalize during resolving disease. Figure 5 shows that during the disease course eosinophil numbers increased and the refractoriness for formyl peptides decreased during amelioration of disease. The data show that normalization of the eosinophil compartment is associated with the improvement of the disease.

## 4. Discussion

The fast and automated point-of-care flow analysis allowed the definition of two separate eosinophil subsets in healthy controls: cells that are responsive to fNLF and those that are unresponsive. fNLF unresponsive cells are CD11b^moderate^ and CD62L^bright^, and do not shift to CD11b^bright^ cells upon activation with fNLF (see Figure 2). The discrete CD11b^bright^ population likely consists of activated cells as part of the eosinophils in the CD11b^moderate^ population move to the CD11b^bright^ population after activation of whole blood with fNLF in vitro. It is, therefore, tempting to speculate that the fNLF unresponsive cells belong to a separate subset of eosinophils. Such a hypothesis is supported by the data published by Mesnil and colleagues, suggesting separate subsets of resident and inflammatory eosinophils [20]. The resident subset was characterized by a high expression of L-selectin (CD62L). Unresponsive CD11b^moderate^ eosinophils might belong to this resident subset as our data show that this population is characterized by a high CD62L expression as well (see Figure 2C). However, future studies should focus on this hypothesis.

We next applied our methodology for eosinophil phenotyping to patients with COVID-19, as eosinophils have been implicated in the pathogenesis of the disease. This is mainly based on the finding that eosinopenia is a hallmark of the disease and the suggestion that the extent of eosinopenia is correlated with the severity of disease [10,11]. However, two recent studies did not report clear eosinopenia, which might point at specific patient characteristics that are associated with low eosinophil counts [21,22]. On the other hand, the eosinophil data in these studies were also consistent with increases in eosinophils during resolution of COVID-19. Our data support the low eosinophil counts in COVID-19 patients at presentation in the hospital but contrast the suggestion that this eosinopenia is specific for infection with SARS-CoV-2 [10,11]. This conclusion is based on the fact that some of the included COVID-19 PCR negative patients suffering from either bacterial or (non-COVID-19) viral infection were also characterized by a low number of eosinophils, albeit less pronounced. Apparently, acute systemic inflammation evoked by infection can lead to eosinopenia irrespective of the micro-organism. This conclusion is strengthened by our previous study showing the challenge of healthy volunteers with lipopolysaccharide (LPS) that is a bacterial microbe-associated molecular pattern (MAMP) leads to acute transient eosinopenia [23]. Together these findings show the surprising result that the eosinophil, historically seen as a typical T2-effector cell, responds to a non-T2 general MAMP in vivo. This supports the view that eosinophils in vivo can be responsive to a non-T2 inflammatory cue.

The eosinophil compartment in the blood of COVID-19 patients is associated with refractoriness for activation with fNLF. This impaired responsiveness is associated with low eosinophil counts. It is, therefore, tempting to speculate that the responsive cells leave the blood and migrate to tissues, leaving behind the refractory cells. This is in line with the idea that inflammatory eosinophils are highly responsive to inflammatory and infectious cues, whereas refractory eosinophils are not. This latter subset might home for the tissues that contain eosinophils in homeostasis [24]. The concept of disappearance of responsive eosinophils from the blood during disease has been put forward in several studies focusing on pathogenesis of allergic asthma [25,26]. This leads to a counterintuitive concept that activation of the eosinophil compartment is associated with the presence of non-responsive cells in the peripheral blood. Support for this type of reasoning is the finding that responsive eosinophils re-emerge in the peripheral blood during improvement of disease (see Figure 5). Importantly, the increase in refractory eosinophils in peripheral blood is not specific for COVID-19 as similar findings were obtained in SARS-CoV-2 PCR negative patients suffering from other infections.

Responsiveness of eosinophils for fNLF was anticipated, as we have described fMLF-induced activation in vitro [18,19]. However, in those old experiments, the magnitude of activation varied considerably between donors. As these data were obtained in cell suspensions with no single cell information, low activation could be interpreted two ways: low responsiveness of all eosinophils, or a low number of highly responsive cells with the majority being non-responsive. Our data support the latter hypothesis. This data also shows that at least part of the eosinophil compartment in humans is responsive to bacterial MAMPs such as fNLF and another part is not. It needs to be emphasized that the percentage of non-responsive eosinophils in COVID-19 is not associated with disease severity (see Figure 4). Together with the finding that low eosinophil counts are already found during very early disease (Figure 3B), the data indicate that the window of opportunity to find differences in the eosinophil compartment is in the COVID-19 population before admission to the hospital. 

Responsiveness for fNLF implies that eosinophils, historically seen as T2-effector cells involved allergic and parasitic diseases, are likely to engage in bacterial infections in humans in vivo as well. This latter hypothesis is strengthened by earlier studies showing the capability of eosinophils to kill bacteria [27] and viruses [28].

These data lead to the essential question whether homing of responsive eosinophils to the tissues in COVID-19 should be prevented? Obviously, this descriptive study will not lead to an answer, but several important issues should be taken into account. Firstly, eosinophils have been shown to kill viruses in vitro through the presence of RNAses present in the granules of these cells [28]. Secondly, we have recently shown that eosinophils in asthma patients have lost their capacity to kill viruses and this is associated with more disease [29]. On the other hand, inflammatory eosinophils present in allergic inflamed tissue have been associated with collateral damage to the host tissues by liberation of granule proteins, such as major basic protein (MBP [30]). The pathogenetic role of eosinophils in at least allergic disease is supported by the finding that inhaled corticosteroids are the key therapy in allergic asthma and is associated with clear suppression of tissue eosinophilia [31]. Therefore, it is still an open issue under which disease conditions eosinophils are guilty by association, play a beneficial role, or play a part in the pathogenesis of the disease. Only the latter situation warrants an intervention targeting these cells.

In summary, the issue whether a specific hyperinflammation in the pathogenesis of COVID-19 is the cause or rather the consequence of infection with the SARS-CoV2 is still a matter of debate. At presentation, COVID-19 patients in our cohort show only very limited signs of a specific systemic inflammation and they cannot be discriminated from patients infected with other micro-organisms or viruses. However, potential differences in co-morbidities in the patient cohorts might have led to confounding data. Moreover, eosinophil counts and eosinophil responsiveness for fNLF did not correlate with disease severity. This implies that the data in this study are associated with severe inflammatory disease *per se*, rather than specific for COVID-19. The lack of disease controls in many COVID-19 studies can lead to wrong interpretations regarding specific immune responses in this disease. In addition, the initiation of disease can still be caused by aberrations in the bradykinin metabolism [3] and coagulopathy [32]. The resulting lung edema and thrombo-embolic events [33] can lead to ischemia and reperfusion injury, leading to tissue damage and release of damage associated molecular patterns (DAMPs) including mitochondrial derived formyl peptides. This can then lead to a toxic triangle of edema, embolisms, and inflammation. In this scenario, inflammation is more the consequence rather than the cause of the disease. The role of eosinophils in the pathogenesis of COVID-19 is uncertain, but targeting these cells is only opportune when it is clear that these cells are not beneficial as virus killing cells.

## Figures and Tables

**Figure 1 cells-10-01109-f001:**
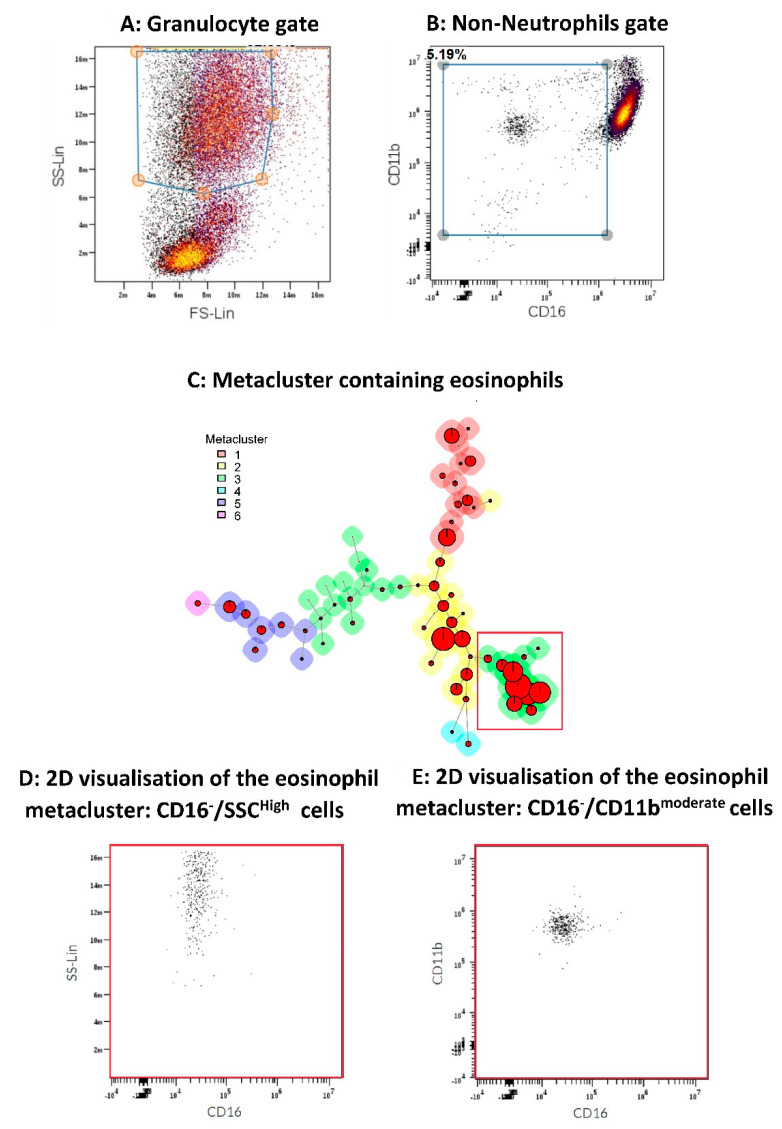
Gating strategy and identification of the eosinophil compartment. (Panel **A**/**B**): gating strategy for obtaining cells in the granulocyte gate (**A**) that are not neutrophils (**B**). Flow cytometric analysis was conducted using the online tool Cytobank (www.cytobank.org; accessed on 7 December 2020). (Panel **C**). Identification of eosinophils (metacluster 3 present in the square) with the use of the FlowSOM algorithm ([14] present in Cytobank). Panels **D**/**E**: characterization of cells in metacluster 3 as being SSC high and CD16 negative (**D**) and CD11b moderate (**E**) eosinophils.

**Figure 2 cells-10-01109-f002:**
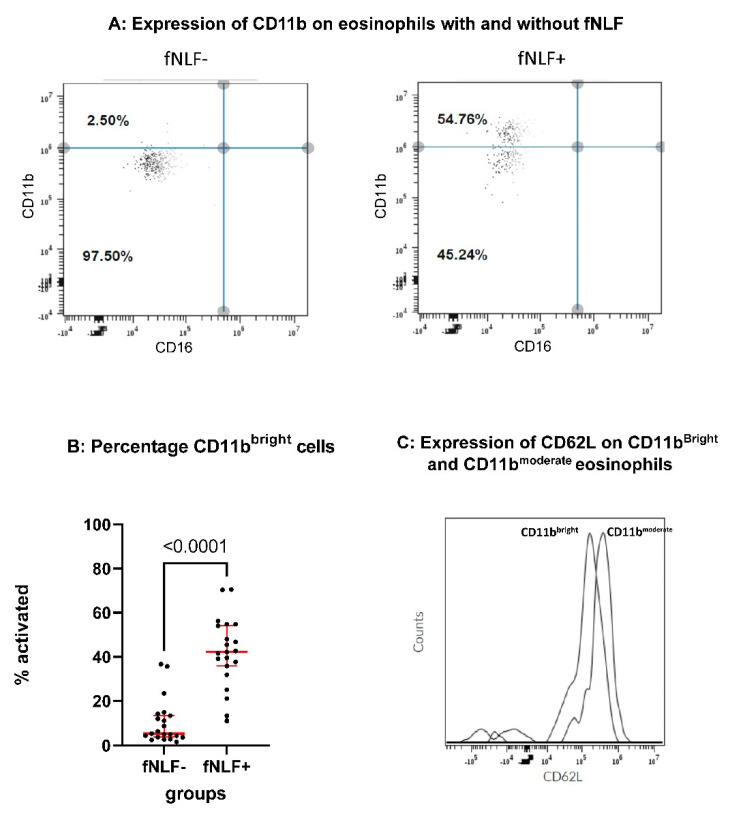
Identification of two types of eosinophils characterized by differences in responsiveness for formyl peptides. (Panel **A**): expression of CD11b on eosinophils from normal healthy controls in the absence of the formyl peptide fNLF. Expression of CD11b (panel **B**) in the presence of fNLF (10 μM). Data are expressed as percentage of CD11b^bright^ cells present in upper left quadrant. CD11b^moderate^ cells are characterized by a high expression of CD62L (panel **C**). Data in (panel **B**) are analyzed using a Wilcoxon matched-pairs signed rank test.

**Figure 3 cells-10-01109-f003:**
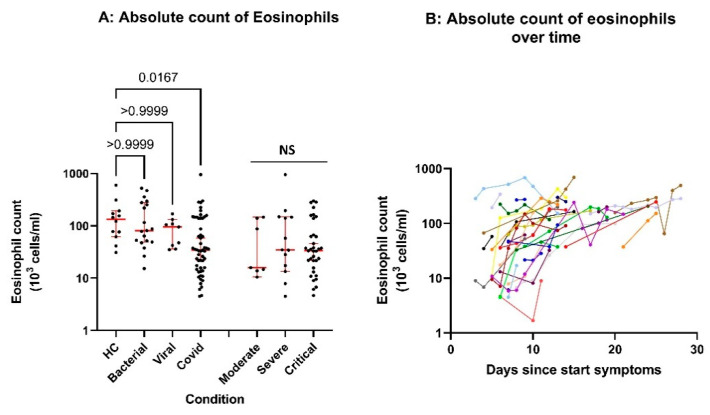
Eosinophil counts under healthy and infectious conditions: (**A**) The absolute number of eosinophils in the blood of different patient groups and controls at admission. Data are expressed as medians +/− confidence interval and statistical significance was tested by Kruskal–Wallis test with Dunn’s test for multiple comparison between groups. (**B**) Blood of COVID-19 patients was drawn at different times after admission to the hospital. The data points of the individual patients are connected, and the different color of the lines represent different patients. At the end of the trajectories the patients were discharged from the hospital.

**Figure 4 cells-10-01109-f004:**
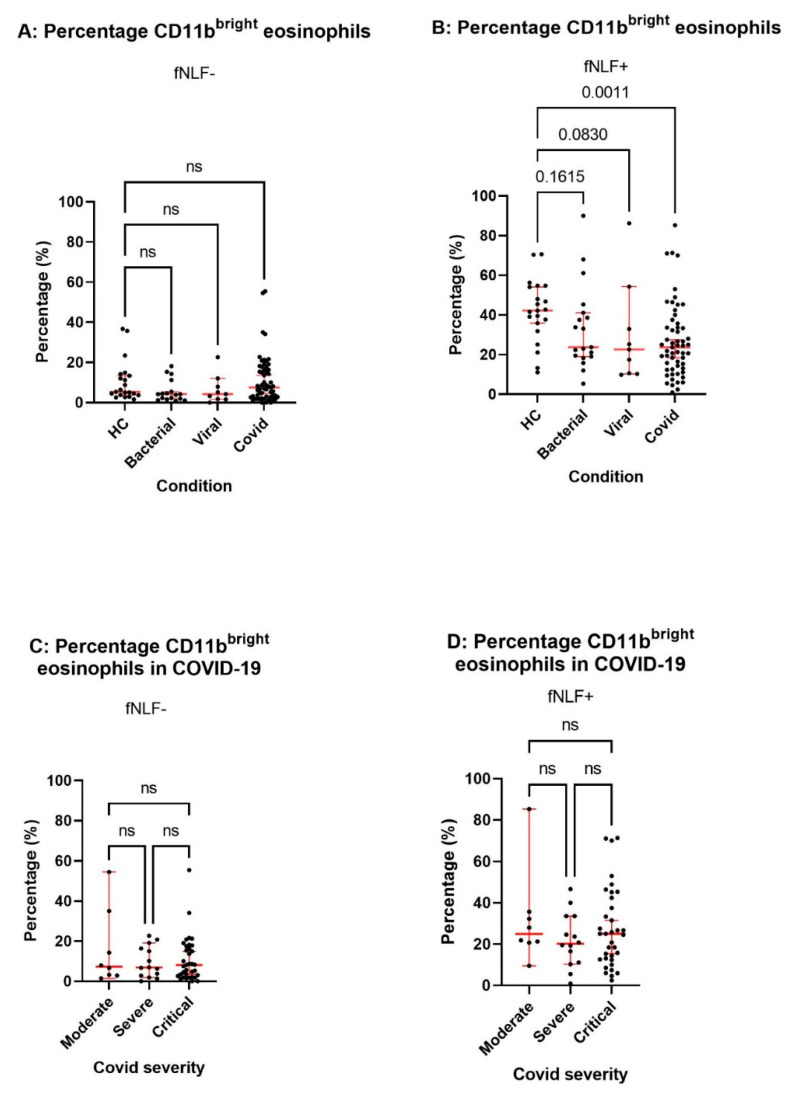
Responsiveness of the eosinophil compartment for formyl peptides in patients with different types of infection. The percentage CD11b^brigth^ eosinophils in the blood of different patient groups and controls at admission in the absence (panel **A**) or presence (panel **B**) of the formyl peptide fNLF (10 μM). The percentage of CD11b^bright^ eosinophils in the blood of COVID-19 patients in the absence (panel **C**) and presence (panel **D**) of fNLF (10 µM). Data are expressed as medians +/− confidence interval and statistical significance is tested by Kruskal–Wallis test with Dunn’s test for multiple comparison between groups.

**Figure 5 cells-10-01109-f005:**
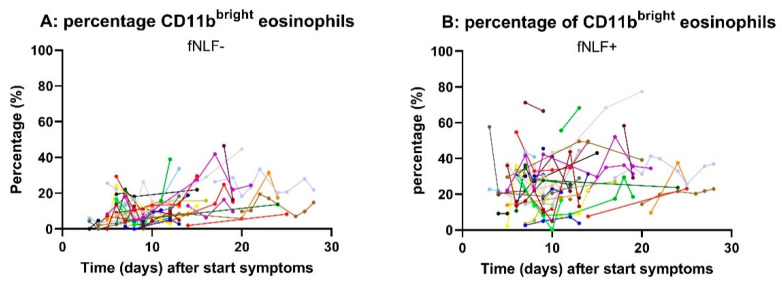
Percentage of CD11b^bright^ eosinophils in the peripheral blood of COVID-19 patients during improvement of the disease. Blood of COVID-19 patients was drawn at different times after admission to the hospital in the absence (panel **A**) or presence (panel **B**) of fNLF (10 μM). The data points of the individual patients are connected, and the different color of the lines represent different patients.

## Data Availability

The data presented in this study are available on request from the corresponding author. The data are not publicly available due to the fact that the data are not completely anonymized.

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
