# Peer review of "The Systemic Immune Response in COVID-19 Is Associated with a Shift to Formyl-Peptide Unresponsive Eosinophils"

_cells, 2021, doi:10.3390/cells10051109_

Round 1
Reviewer 1 Report
I don't feel that the manuscript has significantly improved. There are patient data still missing, which makes it difficult to draw conclusions on the data presented.
Author Response
I don't feel that the manuscript has significantly improved. There are patient data still missing, which makes it difficult to draw conclusions on the data presented.
Answer: the patient data are now presented in the new figure 3A. As can be seen from this figure no statistical difference is present between the three severity groups.
Reviewer 2 Report
The paper named “The Systemic Immune Response in COVID-19 Is Associated with a Shift to Formyl-Peptide Unresponsive Eosinophils” is a interesting descriptive study of the relative eosinopenia found in COVID-19 patients and the refractory behaviour to fNLF activation. This paper only describes the effect but not other study about the possible mechanism or possible application of this effect on COVID-19 patient treatment is presented. However authors give a possible explanation of this effect along the discussion.
Some minor questions are required:
- Have authors any data about if the patients with low eosinophils and a high percentage of refractory behaviour are more prone to severe COVID-19 infection? Or if they stay a more long time in Hospitals. Are there studies of the possible therapy to avoid this behaviour in order to promote the cure of the disease?
- All patients have the same levels of eosinopenia?
- Figure 3B is very interesting however is difficult to understand
- In paragraph 219-220 author talk about figure 3B but seem that sentence is not complete.
- In paragraph 233-234 author say that eosinophils in blood of COVID-19 patients were refractory to activation with fNLF. However in figure 4B the values os p in HC vs covid shows significance. Can author explain this?
- In the same sense as in figure 3B figure 5 is very difficult to understand due to the high amount of information.
- If patients with COVID-19 have eosinopenia with similar behaviour as those infected with virus or bacteria, have authors some explanation to the more severe of COVID-19 Infection. Can this results help to improve the COVID-19 treatment, or to a better classification of COVID-19 patients in order to known how to better manage this patients?
Author Response
Have authors any data about if the patients with low eosinophils and a high percentage of refractory behaviour are more prone to severe COVID-19 infection? Or if they stay a more long time in Hospitals. Are there studies of the possible therapy to avoid this behaviour in order to promote the cure of the disease?
Answer: We have not found any correlation between eosinophils counts and eosinophil responsiveness for fMLF. We have indicated this now in the Discussion section (page 11/lines 343-344)
All patients have the same levels of eosinopenia?
Answer: yes. We have added this extra information to the new figure 3A.
Figure 3B is very interesting however is difficult to understand
Answer: We have clarified this by adding the explanation that at the end of the curves the patients (indicated with their own color) are discharged from the hospital. This information is now added to the result section (page 7 line 223-224).
In paragraph 219-220 author talk about figure 3B but seem that sentence is not complete.
Answer: this is now corrected
In paragraph 233-234 author say that eosinophils in blood of COVID-19 patients were refractory to activation with fNLF. However in figure 4B the values os p in HC vs covid shows significance. Can author explain this?
Answer: the significant difference indicates that the responsiveness of neutrophils towards activation with fMLF is significantly higher in controls compared to COVID-19 patients indicating that COVID19 patients are refractory to activation.
In the same sense as in figure 3B figure 5 is very difficult to understand due to the high amount of information.
Answer: we have now clarified that in figure 3B the individual trajectories are shown of individual patients indicated with their own color. At the end of the curves the patients (indicated with their own color) are discharged from the hospital (see addition of text to the Results section (page7 line 223-225) and the legend of the figure line 214).
If patients with COVID-19 have eosinopenia with similar behaviour as those infected with virus or bacteria, have authors some explanation to the more severe of COVID-19 Infection. Can this results help to improve the COVID-19 treatment, or to a better classification of COVID-19 patients in order to known how to better manage this patients?
Answer: We would have been very excited if our data could be used in staging and/or managing the disease. However, it seems that the systemic effects on eosinophils of SARS-CoV2 infection are general rather than specific. Also there is no correlation between disease severity and eosinophil number (figure 3A) or responsiveness for fNLF (figure 3D). A similar conclusion was drawn for the neutrophil compartment (ref. 8)
Reviewer 3 Report
The authors that examined the innate response in COVID-19 patients, which focused on different eosinopenia populations. They tested the hypothesis that this eosinopenia is COVID-specific and is associated with systemic activation of eosinophils. They found that blood of 15 healthy controls and 75 adult patients with suspected COVID-19, which were included before PCR testing and analyzed by point-of-care automated flow cytometry (CD10, CD11b, CD16 and CD62L) in the absence or presence of a formyl peptide (fNLF). Final, they found that eosinophils in COVID-19 had a similar percentage of CD11bbright cells before activation, but were clearly refractory to activation with fNLF as a much lower percentage of cells end up in the CD11bbright fraction after activation. Low eosinophil numbers in COVID-19 are associated with refractoriness in responsiveness to fNLF. This might be caused by migration of fully functional cells to the tissue.
It is interesting, but should be enhanced during publish.
Some comments as following:
- In the Introduction should enhanced the formyl peptide (fNLF) and its receptor in eosinopenia migration.
- Some markers should be provided the western blot analysis such as CD10, CD11b, CD16 and CD62L
- DAMP molecules should be checked for inflammation induction in patients.
- In the Figure 4, viral sample is based on which one gene or protein should be mention.
Author Response
Some comments as following:
- In the Introduction should enhanced the formyl peptide (fNLF) and its receptor in eosinopenia migration.
Answer: unfortunately, we do not understand this question.
- Some markers should be provided the western blot analysis such as CD10, CD11b, CD16 and CD62L
Answer: it is uncertain whether WB analysis would add much to our findings. Particularly, Mac-1/CD11b is an activation marker that redistributes from the secretory vesicles and specific granules to the plasmamembrane without necessarily differences in total protein. Therefore, these activation markers are so quickly upregulated.
- DAMP molecules should be checked for inflammation induction in patients.
Answer: surely severe COVID19 will liberate DAMP molecules from the affected tissues. However, this is beyond the scope of this study.
- In the Figure 4, viral sample is based on which one gene or protein should be mention.
Answer: this has now been added to the M&M section (page 3, line lines 101-103).
Reviewer 4 Report
Koenderman et al. focus on eosinophils in COVID-19 patients and compare the eosinophil phenotype and activation to other severe infections. The topic of the manuscript is well chosen since eosinophils have historically been understudied in viral infections. The clear strength of the manuscript is the use of cases of other viral and bacterial severe infectionsas controls. The manuscript is well written and results are presented clearly. I have only minor comments:
- M&M section 2.3. the sentence "Aspecific granulocyte activation in the blood…" is repeated twice verbatim in the same chapter.
- line 180 typo in the word eosinophils
- The authors should include and discuss published studies by Rodriguez et al (Cell Reports Medicine, 1, 2020) and Lucas et al (Nature vol 584, 2020) here longitudinal changes in eosinophil immunophenotype have been studied in vivo. These authors did not notice a significant eosinopenia but it could be a results of different patient characteristics and sampling times.
Author Response
M&M section 2.3. the sentence "Aspecific granulocyte activation in the blood…" is repeated twice verbatim in the same chapter.
Answer: this has been corrected.
line 180 typo in the word eosinophils
Answer: this has been corrected.
The authors should include and discuss published studies by Rodriguez et al (Cell Reports Medicine, 1, 2020) and Lucas et al (Nature vol 584, 2020) here longitudinal changes in eosinophil immunophenotype have been studied in vivo. These authors did not notice a significant eosinopenia but it could be a results of different patient characteristics and sampling times.
Answer: initially we decided not to mention these studies as they seemed to add to confusion regarding eosinophils in COVID19 rather than insightful information regarding the role of these cells in COVID19 (no eosinopenia, 12 subtypes of eosinophils based on single cells sequencing that have never been found before, increase of eosinophils with increase disease severity). However, we understand the point of the reviewer and we have now referred to these studies and have added a small discussion to the Discussion Section (see page 9 lines 276-280)
Round 2
Reviewer 1 Report
The authors have improved the manuscript. It is now suitable for publication.
Reviewer 3 Report
No further comment.
This manuscript is a resubmission of an earlier submission. The following is a list of the peer review reports and author responses from that submission.
Round 1
Reviewer 1 Report
The paper by L. Koenderman group describes the systemic immune response in SARS-CoV-2 infected patients. Obtained results are very interesting, and several important issues were drawn. However, the results do not fully answer the main question posed in the manuscript. Nevertheless, the paper is worth studying. The paper deserves publication in the present form.
Reviewer 2 Report
In this study, Koenderman and colleagues describe that patients with COVID-19 present blood eosinopenia and that blood eosinophils are refractory to stimulation with fNLF. This is a purely descriptive study; no mechanisms are presented. Moreover, the authors make a lot of assumptions without showing any evidence for those assumptions.
I have some specific concerns:
- In the abstract, the authors make this assumption: “Low eosinophil numbers in COVID-19 are associated with refractoriness in responsiveness to fNLF, which is likely caused by migration of fully functional cells to the tissue. Such homing might be beneficial as eosinophils have been implicated in viral killing.” However, they don’t show any evidence that the blood eosinophils from COVID-19 patients are functional nor that these eosinophils are able to kill SARS-CoV-2.
- The authors mention that the number of blood eosinophils did not correlate with disease severity; however, they don’t show the data. It would help strengthen the study if the authors show the correlation data.
- Graph 3B is very difficult to follow and understand. If each line represents one patient, it would be better to add a legend and present each patient with a different colour.
- The data showing that blood eosinophils from COVID-19 patients are refractory to fNLF compared to healthy controls is interesting; however, this phenomenon is not specific to COVID-19 as eosinophils from patients with other viral infections and bacterial infection are also refractory. What is the authors’ take on that?
- The authors don’t mention the results presented in figure 4C and 4D.
- The data presented in figure 5A and 5B are not clear (same issue with figure 3B). Also, there are many timepoints missing for some patients and therefore it is very difficult to draw a conclusion from these figures.
- The authors claim that “The data are consistent with the hypothesis that the time needed for recovery mirrors the time of normalization of the eosinophils compartment.” However, I feel that the authors are missing the big picture in the pathogenesis of COVID-19 as many other immunological processes take place during disease and recovery. From their results it is not possible to make that assumption.
Reviewer 3 Report
This manuscript/study looked at the role eosinophils or more specific at the role of Formyl-Peptide (fNLF) unresponsive eosinophils and Covid-19. The authors demonstrate that patients with Covid-19 have lower eosinophil numbers in comparison to control and are associated with refractoriness to fNLF. The authors speculate that this may be due to migration of fully functional cells to the tissue.
This is a nicely done, interesting study with appropriate controls. There are a few minor points outlined below.
Comments:
- In regard to Figure 5, what is the definition and timepoint of “resolution of disease”?